# Behavioral and Electrophysiological Markers of Attention Fluctuations in Children with Hypersomnolence

**DOI:** 10.3390/jcm13175077

**Published:** 2024-08-27

**Authors:** Marine Thieux, Julien Lioret, Romain Bouet, Aurore Guyon, Jean-Philippe Lachaux, Vania Herbillon, Patricia Franco

**Affiliations:** 1Centre de Recherche en Neurosciences de Lyon, INSERM U1028, CNRS UMR5292, Université Claude Bernard Lyon 1, 69500 Lyon, France; ext-julien.lioret@chu-lyon.fr (J.L.); romain.bouet@inserm.fr (R.B.); aurore.guyon@chu-lyon.fr (A.G.); jp.lachaux@inserm.fr (J.-P.L.); vania.herbillon@chu-lyon.fr (V.H.); patricia.franco@chu-lyon.fr (P.F.); 2Unité de Sommeil Pédiatrique, Service d’épileptologie Clinique, des Troubles du Sommeil et de Neurologie Fonctionnelle de l’enfant, Hôpital Femme-Mère-Enfant, Hospices Civils de Lyon, 69500 Lyon, France

**Keywords:** hypoarousal, sleepiness, attention, electrophysiology, pediatric, ADHD, narcolepsy, theta, beta

## Abstract

**Background.** No device is yet available to effectively capture the attentional repercussions of hypersomnolence (HYP). The present study aimed to compare attentional performance of children with HYP, attention deficit hyperactivity disorder (ADHD), and controls using behavioral and electrophysiological (EEG) markers, and to assess their relationship with conventional sleepiness measurements. **Methods.** Children with HYP underwent a multiple sleep latency test (MSLT) and completed the adapted Epworth sleepiness scale (AESS). Along with age-matched children with ADHD, they were submitted to a resting EEG followed by the Bron–Lyon Attention Stability Test (BLAST). The control group only performed the BLAST. Multivariate models compared reaction time (RT), error percentage, BLAST-Intensity, BLAST-Stability, theta activity, and theta/beta ratio between groups. Correlations between these measures and conventional sleepiness measurements were conducted in children with HYP. **Results**. Children with HYP had lower RT and BLAST-Stability than controls but showed no significant difference in BLAST/EEG markers compared to children with ADHD. The AESS was positively correlated with the percentage of errors and negatively with BLAST-Intensity. **Conclusions**. Children with HYP showed impulsivity and attention fluctuations, without difference from children with ADHD for BLAST/EEG markers. The BLAST–EEG protocol could be relevant for the objective assessment of attentional fluctuations related to hypersomnolence.

## 1. Introduction

Sleepiness is a common phenomenon resulting from environmental or pathological conditions (e.g., poor sleep habits or sleep disordered breathing) and can lead to acute or long-lasting consequences (e.g., accidents or school and professional repercussions) [1,2,3]. Sleepiness can be considered as a transitional state: the intrusion of sleep-initiating mechanisms during wakefulness creates an unstable state of vigilance, leading the subject to invest cognitive resources in its stabilization to the detriment of the task at hand [4,5]. When this top-down compensatory effort is effective, performance appears transiently normal, but when it fails, momentary lapses of attention (MLAs) become more frequent, testifying to the emergence of the need for sleep [1,4,6,7,8]. Sleepiness-related MLAs may manifest as impulsivity (i.e., false alarms and excessively reduced reaction time [RT]), inattention (i.e., omissions and massive RT lengthening), or specific mental states (i.e., mind-wandering and mind-blanking) and are more likely to occur in tasks requiring sustained attention (e.g., listening to an explanation or reading a page from a book) [9,10,11,12,13,14].

To date, studies assessing attentional repercussions of sleepiness have yielded heterogenous results. Early studies using sleep deprivation protocols showed that subjects generally achieved normal performance over a short period but failed on tasks requiring more endurance [4]. In addition, patients with central disorders of hypersomnolence commonly have attention complaints that are not always objectively measured [15,16,17]. The difficulty in capturing the attentional repercussions of sleepiness may lie in the tools used. Traditional tests, such as the psychomotor vigilance task (PVT), are well designed to study long lasting drifts away from the ongoing task but are poorly suited to detect MLAs. To explore the multiple dimensions of sleepiness, including neurophysiological changes and behavioral repercussions [18], it is therefore necessary to identify new markers.

Within this context, the present study aimed to compare the attentional performance of children with hypersomnolence (HYP), attention deficit hyperactivity disorder (ADHD) and controls using new markers already known to be impaired in children with ADHD. The secondary objective was to assess the relationship between these markers and conventional sleepiness measurements in children with HYP. At the behavioral level, the assessment of sleepiness-related MLAs can be conducted using the Bron–Lyon Attention Stability Test (BLAST). Using this task, it was previously reported that children with ADHD perform worse than controls [19,20]. At the electrophysiological level, sleepiness can be gauged by the slowing of the waking electroencephalogram (EEG). Theta activity has been positively associated with subjective sleepiness and negatively associated with cognitive performance [21,22,23,24,25,26,27,28,29,30,31]. Moreover, an increase in slow activity and/or a decrease in fast rhythms during rest are linked to sleepiness, cortical hypoarousal, and mind-wandering [32,33,34,35]. This EEG pattern, operationalized through an increased theta/beta ratio (TBR), can serve as the EEG signature of children with ADHD [32,33,36,37,38,39,40]. We hypothesized that children with HYP will show greater attentional impairment compared to controls but without significant difference when compared to children with ADHD and that the behavioral performance on the BLAST and the EEG markers will be associated with conventional measures of sleepiness in children with HYP.

## 2. Materials and Methods

### 2.1. Participants

Thirty-nine children with HYP were referred by their physician for a 48-h hospitalization in the pediatric sleep unit of the Lyon university hospital (France) based on sleep/sleepiness complaints reported by the child, caregivers, or teacher observation. They were included between 2021 and 2022. Forty-one children with ADHD and 53 healthy children were included between 2016 and 2018. The diagnosis of ADHD was confirmed by a neuropsychiatrist based on the DSM-V [41] and using the ADHD Symptom Rating Scale (ADHD-RS) [42]. All children and their legal representatives were informed of the study’s objectives and agreed to participate. The inclusion criteria were: (i) age ≥ 6 and ≤18 years; (ii) regular school curriculum; (iii) absence of major visual, hearing, or upper limb motor impairment; (iv) absence of treatment impacting vigilance levels at the time of the study.

### 2.2. Diagnostic Procedure for Hypersomnolence

Children in the HYP group completed the Adapted Epworth Sleepiness Scale (AESS) [43] which assesses the risk of falling asleep in 8 daily-life situations estimated on a 4-point Likert scale. The total score is the sum of the 8 items: a higher score represents greater sleepiness, and the pathological threshold is set at ≥10. Hospitalization was preceded by 15 days of actimetry and sleep log completion. They also underwent a clinical examination with a pediatric sleep specialist and a polysomnography (PSG) from 8 p.m. to 7 a.m. followed by multiple sleep latency tests (MSLT). The MSLT consisted of 4 or 5 nap opportunities (i.e., 9:00 a.m., 11:00 a.m., 1:00 p.m., 3:00 p.m., and 5:00 p.m.), which were ended after 20 min in the absence of sleep or 15 min after sleep onset. The PSG included 8 electrodes referenced to the mastoids according to the 10–20 system, 2 electro-oculograms, 1 electromyography on the levator menti surface and left and right anterior tibialis muscles, an oral thermistor, thoracic and abdominal belts, an electrocardiogram, a transcutaneous oximeter, and a nasal cannula. The pediatric criteria from the American Academy of Sleep Medicine were used for the visual scoring of sleep stages, arousal, and respiratory events [44].

For the present analyses, children in the HYP group were further classified as either primary or secondary HYP to control for the specificities in sleep characteristics and cognitive profile of children with primary HYP [45]. Primary HYP included patients with narcolepsy, while secondary HYP (2nd HYP) was considered if the children did not meet the diagnostic criteria for idiopathic or secondary narcolepsy, and if another cause could explain the sleepiness. Criteria for the diagnosis of idiopathic narcolepsy were [46]: (i) complaints of excessive daytime sleepiness (EDS) for at least 3 months; (ii) symptoms not better explained by another medical or psychiatric disorder; (iii) absence of secondary narcolepsy; (iv) presence or absence of cataplexy or cerebrospinal fluid hypocretin level < 110 pg/mL; and (v) sleep latency on their MSLT ≤ 8 min and ≥ 2 REM sleep onsets (SOREMP) on their MSLT or PSG.

### 2.3. BLAST Paradigm

The BLAST paradigm was developed to assess the ability to “stay on task” on a second-to-second basis and has been validated in a large cohort of healthy volunteers aged between 6 to 65 years old [19,20]. In the present study, children were told that they would play a game and that they must find a balance between speed and accuracy. The BLAST was presented on a computer screen using Presentation^®^ software. After a few practice trials, the BLAST-Classic session began for around 3 min. A target letter (randomly selected among 12 letters) appeared on the screen (in foveal vision) for 200 ms, then was replaced by a mask (#), and after 300 ms, by a 2-by-2 array of four letters randomly selected from the alphabet. Children had to determine the presence of the initial letter by answering “yes” with their non-dominant hand (or vice versa). The next trial started after 800 ms for a correct response, 4800 ms for an incorrect response, and 3800 ms in case of non-response. Next, 3 BLAST-Color sessions were carried out in which the 4-letter array included a red-letter half the time. In this easier version, the subject had to decide whether one of the letters is red, irrespective of the initial target letter, using the same response system as in the BLAST-Classic version. In both versions, attention abilities were measured using RT in ms, percentage of errors (i.e., false-alarms and omissions), and two specific behavioral measures: BLAST-Intensity and BLAST-Stability, which were created to capture the number and duration of the MLAs. BLAST-Intensity corresponds to the ability to produce long series of fast and correct responses. BLAST-Stability reveals the ability to produce long series of correct responses with a consistent RT, independent of speed. For both measures, the higher the score, the better the attention (for a more detailed description of the measures and psychometric properties of the BLAST, see [19]). The use of both BLAST versions in children with HYP was justified by (i) the fact that an easier (and therefore more “boring”) task (i.e., BLAST-Color) might favor sleepiness-related MLAs, and (ii) the fact that the ADHD and control groups had only performed one or the other. Indeed, while children with HYP performed both BLAST-Classic and BLAST-Color, children with ADHD only performed the BLAST-Classic and controls only performed the BLAST-Color (Figure 1).

### 2.4. Resting-State EEG Recording and Analyses

Before the BLAST, both the HYP and ADHD groups, but not the controls, underwent a resting-state EEG for 2 min with eyes open (EO) and 2 min with eyes closed (EC; Figure 1), using the 10–20 system for electrode placement. EEG data were collected at a sampling frequency of 1024 Hz, with electrode impedances kept below 10 KΩ. Micromed software (v4.04.02, Micromed, Italy) was used for EEG acquisition and was synchronized with the BLAST acquisition computer. Children were seated in front of a computer screen, instructed to be calm, relaxed, and to move as little as possible. During the EO condition, they were asked to focus on a central fixation cross. EEG data from C3 were (pre-)processed using a home-made software [47], including high pass filtering at 1 Hz and automatic rejection of periods compromised by muscle artifacts exceeding a threshold of 400 μV. At least 30% of the signal was retained for each recording, with the first 5 s systematically discarded. A discrete Fourier transform was applied to 8-second time windows with 87.5% overlap. The power spectrum was segmented into frequency bands (δ: 1–4 Hz; θ: 4–8 Hz; α: 8–12 Hz; β: 12–25 Hz), and the theta/beta ratio (TBR) was computed for each window. The final TBR value for each subject was determined by averaging the TBR across all retained epochs. Each recording underwent a visual inspection, and those with insufficient retained signal were excluded from analyses.

### 2.5. Statistical Analysis

Statistical analyses were performed using R software (version 4.0.4) [48]. The statistical significance was set to a *p*-value < 0.05, and the effect sizes were calculated from standard eta squared (η^2^). Missing data were excluded from the analyses. First, a descriptive analysis of sleep and sleepiness characteristics of children with HYP was conducted. Then, type II variance analyses (R-car) were applied to multivariate generalized linear models with a gamma distribution (R-stats). The variables to be explained were the BLAST scores (RT, error %, BLAST-Stability, and BLAST-Intensity) or EEG measures (RTB and θ power). When comparing the age-matched subgroups of HYP vs. control using the BLAST-color, the subgroup was the explanatory variable and the models were adjusted for age—as BLAST performance varies across development [19,20]—and BLAST-Color session number. When comparing the age-matched subgroups of HYP vs. ADHD using the BLAST-Classic or the EEG measures, the subgroup was the explanatory variable, and the models were adjusted for age. Finally, correlations between the BLAST scores (i.e., BLAST-Classic values and mean values of the 3 BLAST-Color sessions) and EEG measures and that of traditional sleepiness measurements (AESS and sleep latency on MSLT) were assessed in children with HYP using Spearman correlations (R-rcorr). All analyses were conducted for the entire HYP group as well as in the 2nd HYP group only.

## 3. Results

Children without BLAST scores (n = 1 with HYP), with an error percentage > 50% (n = 3, 1 with HYP and 2 with ADHD), or a mean RT > 2.5 SD (n = 5 controls) were excluded from the statistical analyses. The final sample was composed of 124 children (37 with HYP, 39 with ADHD, and 48 controls).

### 3.1. Descriptive Analysis

The mean age of the children with HYP was 13 years old (range 6.7–17.8) and 65% were girls. In the group with primary HYP (mean age 12 years old and 57% girls), all children had a diagnosis of idiopathic narcolepsy. The remaining 30 children had 2nd HYP (mean age 13.3 years old and 67% girls) due to an obstructive sleep apnea syndrome (n = 5), delayed sleep phase (n = 8), sleep deprivation (n = 7), psychiatric and/or neurodevelopmental disorders without intellectual deficiency (n = 7), insomnia (n = 2), and restless legs syndrome (n = 1). In the 2nd HYP group, four children had at least one objective criterion for hypersomnia according to conventional measures (i.e., REM sleep latency < 15 min or MSLT sleep latency < 8 min and/or ≥2 SOREM). One of these children had an obstructive sleep apnea on their PSG with an apnea-hypopnea index of 7.5/h. The diagnosis of idiopathic hypersomnia was not retained in the other three children, who presented sleep deprivation measured by actimetry 15 days prior to sleep laboratory evaluation. Sleep and sleepiness data are presented in Table 1.

### 3.2. Group Comparisons

#### 3.2.1. HYP vs. Controls: BLAST-Color

Age-matched subgroups were composed of 34 children with HYP (mean age 12.6 years old, seven with narcolepsy) and 34 controls (mean age 12.7 years old). Adjusted models showed faster RTs (mean: 602 vs. 736 ms, *p* < 0.001, η^2^ = 0.542) and lower BLAST-Stability scores (mean: 32 vs. 40, *p* < 0.001, η^2^ = 0.139) in children in the HYP group compared with controls, respectively. There was no significant difference for the percentage of errors (mean: 3.6 vs. 2.7%, *p* = 0.06, η^2^ = 0.133) or BLAST-Intensity scores (mean: 68 vs. 72, *p* = 0.10, η^2^ = 0.272) (Figure 2). Considering only patients with 2nd HYP, adjusted models showed faster RTs (mean: 597 vs. 736 ms, *p* < 0.001, η^2^ = 0.519) and higher error percentages (mean: 3.9 vs. 2.7%, *p* = 0.01, η^2^ = 0.170) in children with 2nd HYP compared to controls. Adjusted models also showed an interaction between group and age for BLAST-Stability (*p* = 0.04, η^2^ = 0.158) and BLAST-Intensity (*p* = 0.001, η^2^ = 0.316): children with 2nd HYP showed lower BLAST-Stability (mean: 32 vs. 40) and BLAST-Intensity (mean: 68 vs. 72) compared to controls but this effect was less pronounced as age increased.

#### 3.2.2. HYP vs. ADHD: BLAST-Classic and EEG

The age-matched subgroups consisted of 23 children with HYP (mean age 10.9 years old, four with narcolepsy) and 23 children with ADHD (mean age 10.7 years old). Among the latter, 12 children had the inattention subtype, 1 had the hyperactivity subtype, and 10 had the mixed subtype (i.e., inattention and hyperactivity). Adjusted models showed no significant difference between children with HYP and ADHD for RT (mean: 931 vs. 1000 ms, *p* = 0.36, η^2^ = 0.406), error percentage (mean: 16.4 vs. 15.6%, *p* = 0.73, η^2^ = 0.127), BLAST-Stability (mean: 15 vs. 18, *p* = 0.42, η^2^ = 0.066), and BLAST-Intensity (mean: 33 vs. 34, *p* = 0.74, η^2^ = 0.171). These results remained comparable when considering only patients with 2nd HYP (mean RT: 880 vs. 1000 ms, *p* = 0.24, η^2^ = 0.377; mean error %: 15.2 vs. 15.6%, *p* = 0.78, η^2^ = 0.122; mean BLAST-Stability: 16 vs. 18, *p* = 0.44, η^2^ = 0.062; and mean BLAST-Intensity: 35 vs. 34, *p* = 0.76, η^2^ = 0.154).

Regarding EEG, 20 children with ADHD and 15 children with HYP (3 with narcolepsy) had CE available data, while 19 children with ADHD and 12 children with HYP (2 with narcolepsy) had OE available data. The groups were not significantly different in terms of age (*p* > 0.05). Adjusted models showed no significant difference between children with HYP and ADHD for both the CE and OE EEG markers: CE RTB (mean: 5.36 vs. 5.34, *p* = 0.78, η^2^ = 0.246), CE θ power (mean: 37.43 vs. 36.89, *p* = 0.48, η^2^ = 0.253), OE RTB (mean: 5.40 vs. 5.35, *p* = 0.55, η^2^ = 0.140), and OE θ power (mean: 37.47 vs. 36.37, *p* = 0.26, η^2^ = 0.202). These results remained comparable when considering only patients with 2nd HYP vs. ADHD (CE mean RTB: 5.30 vs. 5.34, *p* = 0.80, η^2^ = 0.215; CE mean θ power: 37.20 vs. 36.89, *p* = 0.45, η^2^ = 0.262; OE mean RTB: 5.36 vs. 5.35, *p* = 0.77, η^2^ = 0.111; and OE mean θ power: 37.23 vs. 36.37, *p* = 0.32, η^2^ = 0.179).

### 3.3. Correlations between BLAST, EEG Parameters, and Sleepiness in Children with HYP

In both the total HYP and 2nd HYP groups, the AESS score was correlated with the percentage of errors (rho = 0.33, *p* = 0.04, and rho = 0.44, *p* = 0.01 respectively) and the BLAST-Classic Intensity score (rho = −0.34, *p* = 0.04, and rho = −0.41, *p* = 0.03 respectively): the higher the sleepiness, the higher the percentage of errors and the lower the BLAST-Intensity score. There was no other significant correlation between BLAST-scores or EEG activity and conventional measures of sleepiness.

## 4. Discussion

According to our first hypothesis, children with HYP had comparable attentional performances to children with ADHD when using BLAST-Classic behavioral and EEG markers. Furthermore, when compared to controls, children with HYP showed signs of impulsivity and attentional fluctuations characterized by faster BLAST-Color RT and lower BLAST-Stability. While these findings emphasize the need for high temporal resolution tools to effectively capture MLAs that disrupt long-term attention stability; they also illustrate the overlap between sleepiness and attention disorders.

In adults, over 60% of people with central hypersomnia have clinically relevant attention complaints. In addition, EDS is reported by over 45% of patients with ADHD [49]. As seen in children [50,51], this overlap could stem from an altered vigilance regulation, with some authors proposing a “narcolepsy-like” ADHD subtype characterized by cortical hypoarousal [52,53,54]. The attempt to up-regulate vigilance by implementing compensatory mechanisms, such as redirecting cognitive resources from the ongoing task or engaging in self-stabilizing behaviors (e.g., motor-restlessness, talkativeness, fidgeting), may explain both inattentive and hyperactive manifestations [53,55]. This “unstable state” hypothesis [4] is supported by several studies using sleep-restriction/deprivation protocols. For example, healthy sleep-restricted children showed auto-stabilization behaviors and attention disorders for complex tasks only, suggesting an efficient compensation for easier ones [56,57,58,59]. This hypothesis is also supported by neurophysiological evidence. According to Posner and Petersen, attention efficiency depends on the alert system, which corresponds to structures involved in vigilance regulation [60] and contributes both to the optimal activation of relevant regions for the task at hand (i.e., task-positive network) and to the deactivation of irrelevant regions (i.e., default-mode network [DMN]) [61]. Accordingly, hypovigilance would disrupt this functional balance, creating the typical brain state of MLA occurrence [14] and impacting BLAST performance [19]. Indeed, attention deficits following sleep perturbation have been associated with a reduced activation of the task-positive fronto–parietal network and/or a reduced deactivation of DMN regions [62,63]. An aberrant functioning of these networks has also been reported in patients with ADHD [64,65], central hypersomnia [66,67,68], and chronic sleepiness [69,70]. The θ activity and TBR could be the electrophysiological signature of this network’s imbalance [34,35]. In line with our hypothesis, children with ADHD and HYP showed comparable resting EEG activity, making θ activity and RTB relevant markers of vigilance disturbance. The EEG slowing before (and during) attention tasks has been previously associated with performance impairment in healthy sleep-deprived adults [23,30] and subjects with ADHD [71]. These changes in slow-frequency oscillations are mainly related to isolated sleep-like slow-waves [29], which, depending on their localization, could explain both impulsivity and inattention [12,13,30]. As in slow-wave sleep, where slow waves are associated with a modulation of connectivity [72], their increase during wakefulness is linked to a general reduction in functional connectivity [30]. While these local “sleep islands”, associated with brain activity modulations and cognitive and behavioral alterations, could be an interesting neural mechanism for understanding attentional repercussions of dysregulated vigilance; direct evidence of such mechanisms remains to be uncovered.

Regarding the validation of the new markers of HYP proposed herein, our second hypothesis is partly validated: attentional performances assessed with a particularly short task (<3 min) are correlated to certain clinical characteristics of hypersomnolence. Indeed, the AESS was correlated with the error percentage and BLAST-Intensity scores of the BLAST-Classic. However, other BLAST performance or EEG markers were not correlated with subjective nor objective conventional sleepiness measures in children with HYP, as often reported in the literature [10]. In the case of sleep deprivation conditions or in patients with narcolepsy, cognitive performance does not vary or correlate (or only slightly) with self-reported sleepiness (i.e., assessed via the Epworth sleepiness, the Stanford sleepiness, or analog scales) [17,73,74,75,76]. Regarding objective assessment of sleepiness using the MSLTs, results are contradictory. Some studies show an absence of variation (and correlation) between performance and sleepiness [74,75,77] while others show an increase in executive and attentional disorders with a decrease in sleep latency and an increase in SOREMP [16,78]. These results reaffirm the limitations of scales in conditions where the subject’s response abilities could be affected by sleepiness, and of certain objective measures in discriminating high levels of sleepiness, thus re-emphasizing the limitations of these conventional measures in predicting every sleepiness’ dimension and repercussion, as well as highlighting the clinical interest of the BLAST.

The present study has some limitations. First, since this research was conducted in a clinical setting, this led to a certain diagnostic heterogeneity in children with HYP. Although the results underline the value of the BLAST–EEG paradigm for the assessment of the attention impact of hypersomnolence of various etiologies, the attentional profile of children with narcolepsy may be slightly different, as suggested by previous studies [15,79]. Future studies examining behavioral and EEG markers in a larger group of children with narcolepsy only and in children according to their ADHD subtype would therefore be necessary. Moreover, since EEG markers could not be studied in healthy children, further investigations are required to determine their pathological characterization in children with HYP and ADHD. Second, in children with ADHD and controls, attentional performance may have been influenced by variables not controlled here (e.g., habitual sleep/wake pattern, PSG parameters the night before the BLAST, physical activities, anxiety, or depression). These aspects could be evaluated in a future study to refine our understanding of the mechanisms underlying the relationship between sleep/sleepiness and attention disorders. Third, while the BLAST paradigm appears to be relevant to capture MLAs related to local and transient vigilance instability, attention fluctuations related to sleep perturbations are classically studied using the PVT which shows sensitivity to circadian processes, sleep pressure, and agents influencing vigilance. Studies are needed to assess the sensitivity of BLAST to these parameters. Of note, the BLAST–EEG was herein conducted at 4 p.m. to control for the potential influence of the time of the day on attention performance. Finally, methodological considerations need to be considered regarding EEG markers. First, the analysis was conducted on fixed frequency bands, but the use of individually selected bands based on the alpha peak [37] together with 1/f weighting, to account for interindividual variability, might be preferred. Second, to facilitate the recording in pediatric population with ADHD and/or HYP, which present with their own specificities (e.g., agitation or drowsiness), the recording duration was set to 2 min in each condition (vs a median of 5 min in the literature [40]). However, a longer recording period could lead to a better stability of the TBR. Third, the validity of the TBR might be better in the CE condition, as the OE condition may be contaminated by eye (micro)movements—particularly in children with ADHD—that may go unnoticed during signal cleaning. Finally, other EEG markers could be relevant to assess the impact of sleepiness, such as α activity or θ/α ratio during the resting-state EEG [80], or the occurrence of local slow-waves during the BLAST [13].

## 5. Conclusions

While underlying the relationship between sleepiness and attention disorders, this study emphasizes the interest of new markers to explore multiple dimensions of sleepiness and its repercussions. Because of its ability to capture brief attentional lapses, the BLAST–EEG protocol constitutes a promising tool for the rapid and objective assessment of attentional fluctuations in subjects with attention and sleepiness complaints.

## Figures and Tables

**Figure 1 jcm-13-05077-f001:**
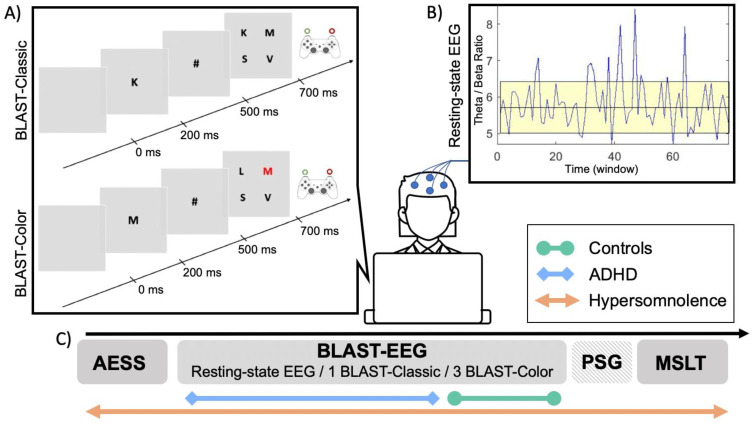
Experimental design. (**A**) One trial of the BLAST-Classic (top) and BLAST-Color (bottom). In each trial, a target letter appeared on the screen, followed by a mask, and then by a 2-by-2 array of four letters. Children were instructed to detect the presence or the absence of the target letter (BLAST-Classic) or of a red letter (BLAST-Color) in the array. Responses were given manually with a joystick (i.e., “yes” responses were given using the non-dominant hand; “no” responses were provided using the dominant hand). For each BLAST session, the total testing time was around 3 min, with 60 trials to complete. (**B**) The resting-state EEG was recorded before the BLAST for 2 min with opened eyes and 2 min with closed eyes, to assess the theta and beta activity, as well as the theta/beta ratio (TBR). Temporal variations of the TBR are represented here for the open eye condition. (**C**) Children with hypersomnolence (orange) completed the entire protocol; children with ADHD (blue) completed a resting EEG and 1 session of BLAST-Classic; control children (green) completed 3 sessions of BLAST-Color. AESS: adapted Epworth Sleepiness Scale; BLAST: Bron–Lyon Attention Stability Test; PSG: polysomnography; and MSLT: multiple sleep latency tests.

**Figure 2 jcm-13-05077-f002:**
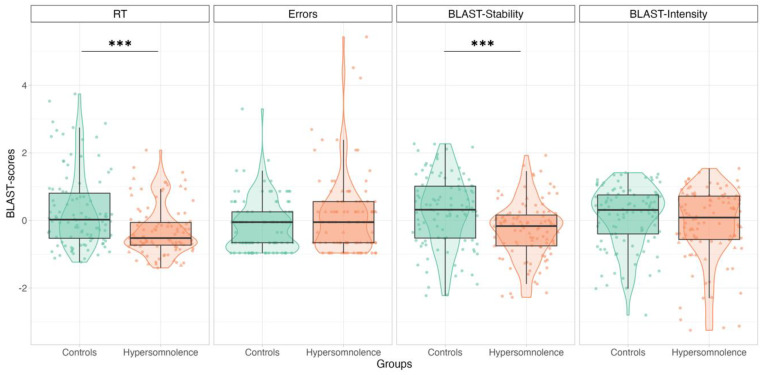
Between-group comparisons of BLAST-Color performance. RTs (ms), errors (%), BLAST-Stability scores, and BLAST-Intensity scores in controls (green) and children with HYP (orange). Each dot represents a participant’s score for the three BLAST-Color sessions. The central line of each boxplot corresponds to the median, and the upper and lower parts of the box to the first and third quartiles. Children with primary HYP are represented by triangles and those with 2nd HYP are represented by dots. Significant differences are represented by stars: ***: *p* < 0.001 (R-ggplot2).

**Table 1 jcm-13-05077-t001:** Sleep and sleepiness characteristics of children with hypersomnolence.

	Total HYP (n = 37)	Primary HYP (n = 7)	Secondary HYP (n = 30)
Sleep characteristics—PSG			
TST, min	539 (393–628)	562 (533–627)	534 (393–628)
Sleep latency, min	13.1 (0–146)	0.2 (0–2.3)	19.5 (0–146)
REM latency, min	127 (0–272)	2 (0–68)	142 (1.5–271.5)
REM latency < 15 min, n	19 (7)	71 (5)	7 (2)
Sleep efficiency, %	91 (73–99)	91 (88–98)	91 (73–99)
N1, %	6.8 (1.4–15.7)	10.3 (3.5–15.6)	6.7 (1.4–15.7)
N2, %	53.6 (29.2–68)	43.7 (29.2–64.7)	53.7 (38.3–68)
N3, %	17.4 (6.8–31.8)	17.4 (7.2–31.8)	17.3 (6.8–31)
REM, %	22.4 (10.6–40.8)	24.6 (21.1–40.8)	21.8 (10.6–32.8)
Arousals and micro-arousals index, n/h	11.8 (6.2–45)	12.4 (9.2–30.9)	10.8 (6.2–45)
Sleepiness characteristics			
AESS	13 (0–24)	20 (16–24)	12 (0–22)
Pathological AESS, n	73 (27)	100 (7)	67 (20)
MSLT sleep latency, min	15.8 (2.3–20)	4.5 (2.3–13.9)	17 (3.6–20)
MSLT sleep latency < 8 min, n	19 (7)	71 (5)	7 (2)
MSLT SOREMP, %	0 (0–100)	75 (25–100)	0 (0–50)

Note. All values in columns 2, 3, and 4 are reported as median (range) or % (n). Abbreviations: AESS: adapted Epworth Sleepiness Scale; MSLT: multiple sleep latency test; N1: stage 1 sleep; N2: stage 2 sleep; N3: stage 3 sleep; PSG: polysomnography; SOREMP: sleep onset REM; TST: total sleep time.

## Data Availability

Data may be available on request from the corresponding author.

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
