# Peer review of "Behavioral and Electrophysiological Markers of Attention Fluctuations in Children with Hypersomnolence"

_jcm, 2024, doi:10.3390/jcm13175077_

Round 1

Reviewer 1 Report

Comments and Suggestions for Authors

The topic of hypersomnia in children and adults is a fairly common problem that can significantly reduce the academic performance of schoolchildren and students. In this regard, the topic of the manuscript is relevant and interesting for practical neurology and somnology. The manuscript does not contain data on the sleep-rest regime in the observed groups (going to bed, waking up time, independent or by alarm clock waking up), and also raises the question of the employment of the observed groups - only academic workload or is there additional sports load, this also affects sleep. How did the researchers select the healthy control group? Were any tests conducted and what additional research methods were used? Therefore, the question of health in terms of sleep is very conditional. Also, the question arose whether a questionnaire was conducted on anxiety and depression, their presence can also affect the quality and quantity of sleep. If all these studies were conducted, then it makes sense to include their results in the manuscript. Table 1 shows the testing results of only the group with hypersomnia, for a more accurate analysis, it makes sense to add the testing results of the other two groups. This is important for a more correct analysis of the BLAST test results.

Author Response

Response to Reviewer 1 Comments

The topic of hypersomnia in children and adults is a fairly common problem that can significantly reduce the academic performance of schoolchildren and students. In this regard, the topic of the manuscript is relevant and interesting for practical neurology and somnology.

We thank the reviewer for their positive comments and feedback on our work. Please find the detailed responses below (in blue) and the corresponding revisions/corrections in the re-submitted files (in red). We hope that this new and improved version of the manuscript will be suitable for publication. 

The manuscript does not contain data on the sleep-rest regime in the observed groups (going to bed, waking up time, independent or by alarm clock waking up), and also raises the question of the employment of the observed groups - only academic workload or is there additional sports load, this also affects sleep.

We agree with the reviewer that the sleep-wake rhythm, as well as workload and other, can have a significant impact on sleep and sleepiness. As mentioned in the method section, all three subgroups were composed of children following a regular school curriculum, and none of them were employed.

p.5 The inclusion criteria were: … (ii) regular school curriculum”,

Controls and children with ADHD were screened for sleep disturbance the night before the BLAST, to ensure their ability to perform the task. However, data on sleep-rest regime or physical activity were only available in the group with HYP. To limit the number of variables not shared between the 3 groups, we decided to not include them in the present manuscript. We consider this to be an important limitation and have added a sentence in the limitation section (see last point below).

How did the researchers select the healthy control group? Were any tests conducted and what additional research methods were used? Therefore, the question of health in terms of sleep is very conditional.

We agree with the reviewer that the increase in poor sleep hygiene in the general population raises questions about the existence of “healthy controls” in terms of sleep. Healthy children were included outside the clinical setting to obtain normative BLAST data. They had no complaints about sleep or sleepiness, but no further tests were carried out on their sleep-wake parameters.

Also, the question arose whether a questionnaire was conducted on anxiety and depression, their presence can also affect the quality and quantity of sleep. If all these studies were conducted, then it makes sense to include their results in the manuscript.

As before, these data were only available for the group with HYP, which underwent a full clinical evaluation (i.e. anxiety and depression questionnaires, meeting with a psychiatrist if necessary). To focus on the shared variables between the 3 groups, we decided to not include them in the present manuscript. We have added a sentence in the limitation section (see last point below). Of note, the number of children with 2nd HYP with anxiety / depression is reported in the descriptive analysis section.

p.10 “The remaining 30 children had 2nd HYP (mean age 13.3 years old, 67% girls) due to … psychiatric and / or neurodevelopmental disorders without intellectual deficiency (n = 7)”.

Table 1 shows the testing results of only the group with hypersomnia, for a more accurate analysis, it makes sense to add the testing results of the other two groups. This is important for a more correct analysis of the BLAST test results.

We agree with the reviewer regarding the importance of objective sleep measures and sleepiness parameters for the analysis of BLAST results, particularly in children with ADHD who are known to suffer from various sleep disturbances. However, due to the design of the study, these variables were not assessed in healthy children and those with ADHD. All these points constitute a major limitation of our study; we have added a section including all missing data in the limitations section.  

p.15 “…, in children with ADHD and controls, attentional performance may have been influenced by variable not controlled here (e.g. habitual sleep/wake pattern, PSG parameters the night before BLAST, physical activities, anxiety or depression). These aspects could be evaluated in a future study to refine our understanding of the mechanisms underlying the relationship between sleep/sleepiness and attention disorders.”

Reviewer 2 Report

Comments and Suggestions for Authors

This study compared the attentional performance of children with hypersomnolence, attention deficit hyperactivity disorder (ADHD), and controls using both behavioral and electrophysiological markers. Additionally, the relationship between these markers and conventional sleepiness measurements was examined. The authors found that children with hypersomnolence had lower reaction time and Bron-Lyon Attention Stability Test (Blast) stability compared to controls. However there were no significant differences in Blast or EEG markers between children with hypersomnolence and those with ADHD. In addition, the adapted Epworth Sleepiness scale (AESS) score was positively correlated with the percentage of errors and negatively correlated with BLAST intensity. The topic is interesting because attention, as a cognitive function has not been extensively studied in the context of excessive daytime sleepiness. Overall, the manuscript is well written, with excellent coherence between its various sections from the INTRODUCTION to the CONCLUSION. Below are two minor comments:

Materials and methods

- Page 5; 2.1: The study population which includes patients with hypersomnolence, patients with ADHD and healthy children is clearly defined. However, could you specify please how hypersomnia was initially identified in the relevant sub-group? was it reported by the children themselves, their parents or guardians, or throw some other means?

Results

- Page 10 ; 3.1 : According to the “Materials and Methods”, 39 children were initially included based on complaints of hypersomnia, which may be subjective. Was this complaint confirmed by polysomnographic and multiple sleep latency test (MSLT) studies in all the 37 patients who were ultimately retained in the Hypersomnia sub-group.

Author Response

Response to Reviewer 2 Comments

This study compared the attentional performance of children with hypersomnolence, attention deficit hyperactivity disorder (ADHD), and controls using both behavioral and electrophysiological markers. Additionally, the relationship between these markers and conventional sleepiness measurements was examined. The authors found that children with hypersomnolence had lower reaction time and Bron-Lyon Attention Stability Test (Blast) stability compared to controls. However there were no significant differences in Blast or EEG markers between children with hypersomnolence and those with ADHD. In addition, the adapted Epworth Sleepiness scale (AESS) score was positively correlated with the percentage of errors and negatively correlated with BLAST intensity. The topic is interesting because attention, as a cognitive function has not been extensively studied in the context of excessive daytime sleepiness. Overall, the manuscript is well written, with excellent coherence between its various sections from the INTRODUCTION to the CONCLUSION.

We thank the reviewer for their positive comments and feedback on our work. Please find the detailed responses below (in blue) and the corresponding revisions/corrections in red in the re-submitted files. We hope that this new and improved version of the manuscript will be suitable for publication. 

Materials and methods

Page 5; 2.1: The study population which includes patients with hypersomnolence, patients with ADHD and healthy children is clearly defined. However, could you specify please how hypersomnia was initially identified in the relevant sub-group? was it reported by the children themselves, their parents or guardians, or throw some other means?

Children with HYP were addressed for a 48-hour hospitalization in the pediatric sleep unit of the Lyon university hospital (France) by their physician, based on sleep / sleepiness complaint reported by the child, caregivers, or observation by the teacher. We added this useful information in the manuscript.

p.5 “Thirty-nine children with HYP were referred by their physician for a 48-hour hospitalization in the pediatric sleep unit of the Lyon university hospital (France), based on sleep / sleepiness complaints reported by the child, caregivers, or teacher observation. They were included between 2021 and 2022.

Results

Page 10 ; 3.1 : According to the “Materials and Methods”, 39 children were initially included based on complaints of hypersomnia, which may be subjective. Was this complaint confirmed by polysomnographic and multiple sleep latency test (MSLT) studies in all the 37 patients who were ultimately retained in the Hypersomnia sub-group.

We thank the reviewer for this remark. Children with primary HYP had pathological criteria for narcolepsy using objective measures (sleep latency and SOREMP). Most other children with a complaint of hypersomnolence did not meet the criteria of hypersomnia. Pathological thresholds for MSLT sleep latency are already reported in Table 1. We have added a line in the table for PSG REM sleep latency < 15 minutes and a sentence reporting that 4 children among the 30 with 2nd HYP who ultimately showed at least one objective criterion of hypersomnia (PSG REM latency <15 min or MSLT sleep latency < 8 min or/and > 2 SOREM).

p.10 “In the 2nd HYP group, 4 children had at least one objective criterion for hypersomnia according to conventional measures (i.e. REM sleep latency < 15 min or MSLT sleep latency < 8 min and/or > 2 SOREM). One of these children had an obstructive sleep apnea on PSG with an apnea-hypopnea index of 7.5/h. The diagnosis of idiopathic hypersomnia was not retained in the other 3 children, who presented sleep deprivation measured by actimetry 15 days prior to sleep laboratory evaluation.